# Comparison of Characteristics and Outcomes of Multisystem Inflammatory Syndrome, Kawasaki Disease and Toxic Shock Syndrome in Children

**DOI:** 10.3390/medicina59030626

**Published:** 2023-03-21

**Authors:** Lizete Klavina, Liene Smane, Anda Kivite-Urtane, Lauma Vasilevska, Zane Davidsone, Emils Smitins, Dace Gardovska, Inguna Lubaua, Ieva Roge, Zanda Pucuka, Anija Meiere, Jana Pavare

**Affiliations:** 1Department of Continuing Education, Riga Stradins University, Children’s Clinical University Hospital, LV-1004 Riga, Latvia; 2Department of Pediatrics, Riga Stradins University, Children’s Clinical University Hospital, LV-1007 Riga, Latvia; liene.smane@bkus.lv (L.S.); jana.pavare@bkus.lv (J.P.); 3Department of Public Health and Epidemiology, Institute of Public Health, Riga Stradins University, LV-1010 Riga, Latvia

**Keywords:** SARS-CoV-2, MIS-C, Kawasaki disease, TSS, pediatrics

## Abstract

*Background and Objectives:* Since the first cases of multisystem inflammatory syndrome in children (MIS-C) in April 2020, the diagnostic challenge has been to recognize this syndrome and to differentiate it from other clinically similar pathologies such as Kawasaki disease (KD) and toxic shock syndrome (TSS). Our objective is to compare clinical signs, laboratory data and instrumental investigations between patients with MIS-C, KD and TSS. *Materials and Methods:* This retrospective observational study was conducted at the Children’s Clinical University Hospital, Latvia (CCUH). We collected data from all pediatric patients <18 years of age, who met the Centers for Disease Control and Prevention case definition for MIS-C, and who presented to CCUH between December 2020 and December 2021. We also retrospectively reviewed data from inpatient medical records of patients <18 years of age diagnosed as having KD and TSS at CCUH between December 2015 and December 2021. *Results*: In total, 81 patients were included in this study: 39 (48.1%) with KD, 29 (35.8%) with MIS-C and 13 (16.1%) with TSS. In comparison with TSS and KD, patients with MIS-C more often presented with gastrointestinal symptoms (abdominal pain (*p* < 0.001), diarrhea (*p* = 0.003)), shortness of breath (*p* < 0.02) and headache (*p* < 0.003). All MIS-C patients had cardiovascular involvement and 93.1% of MIS-C patients fulfilled KD criteria, showing higher prevalence than in other research. Patients with KD had higher prevalence of cervical lymphadenopathy (*p* < 0.006) and arthralgias (*p* < 0.001). In comparison with KD and TSS, MIS-C patients had higher levels of ferritin (*p* < 0.001), fibrinogen (*p* = 0.04) and cardiac biomarkers, but lower levels of platelets and lymphocytes (*p* < 0.001). KD patients tended to have lower peak C-reactive protein (CRP) (*p* < 0.001), but higher levels of platelets. Acute kidney injury was more often observed in TSS patients (*p* = 0.01). Pathological changes in electrocardiography (ECG) and echocardiography were significantly more often observed in MIS-C patients (*p* < 0.001). *Conclusions:* This research shows that MIS-C, KD and TSS have several clinical similarities and additional investigations are required for reaching final diagnosis. All the patients with suspected MIS-C diagnosis should be examined for possible cardiovascular involvement including cardiac biomarkers, ECG and echocardiography.

## 1. Introduction

In April 2020, several reports from the United Kingdom documented presentation of symptoms in children similar to those of Kawasaki disease (KD) or Toxic shock syndrome (TSS). Since then, there have been similar reports in other parts of the world [1,2,3]. This condition has later been termed Multisystem Inflammatory Syndrome in children (MIS-C), and has been defined by the CDC [4]. According to the case definition, affected children or adolescents <21 years old have fever ≥38.0 C at least 24 h, multisystem (≥2 organ) involvement (cardiac, renal, respiratory, hematologic, gastrointestinal, dermatologic or neurological) and at least one laboratory feature suggesting MIS-C, evidence of clinically severe illness requiring hospitalization and a history of COVID-19 disease (positive for current or recent SARS-CoV-2 infection by RT-PCR, serology, or antigen test; or exposure to a suspected or confirmed COVID-19 case within the 4 weeks prior to the onset of symptoms). Other possible diagnoses must be excluded [4]. This case definition was used in this research for MIS-C patient selection, but since January 2023 case definition has been updated: clinical criteria are specified, requesting CRP level ≥30 mg/L, and at least two confirmed signs or organ involvement from the following: cardiac involvement, mucocutaneous involvement, shock, gastrointestinal involvement and hematologic involvement. Linkage to COVID-19 (laboratory approved or epidemiological data) is now estimated 60 days prior to hospitalization [5].

Distinguishing MIS-C from other hyperinflammatory conditions such as Kawasaki disease and toxic shock syndrome can be challenging for health care providers. Therefore, displaying the major clinical and laboratory differences of these pathologies is critical to making a proper differential diagnosis.

At the beginning of the COVID-19 pandemic, the first MIS-C cases were divided into different phenotypes depending on clinical presentation—Kawasaki disease-like, toxic shock-like, and also viral myocarditis according to some authors [6]. Now, MIS-C diagnosis contains a spectrum of clinical manifestations from a mild to a severe course of the disease, and is classified as follows:(1)MIS-C without overlap with Kawasaki disease (KD) or acute COVID-19 (35% of cases)—these patients commonly present with shock, cardiac dysfunction, gastrointestinal involvement. and have significantly high levels of CRP and ferritin.(2)MIS-C overlapping with KD (35% of cases)—these patients are on average younger than in the other two groups (median age 6 years), and have Kawasaki syndrome-like symptoms—rash, mucocutaneous involvement, and less commonly shock and myocardial dysfunction. Multiple case series report that about 40–50% of children with MIS-C meet the criteria for complete or incomplete KD.(3)MIS-C overlapping with severe acute COVID-19 (30% of cases)—these patients typically present with respiratory system involvement, including pneumonia and acute respiratory distress syndrome. Most have positive SARS-CoV-2 PCR at the time of presentation; they also tend to be older and more often have comorbidities.

The highest mortality rate is observed in the subgroup overlapping with severe acute COVID-19. It is important to emphasize that coronary artery lesions are observed equally in all the subgroups of MIS-C [7].

Notably, given the clinical overlap with KD, for many MIS-C patients the only differentiating feature may be the evidence of prior COVID-19 infection. However, as the pandemic has progressed and a greater proportion of the population has had prior SARS-CoV-2 infection and/or been vaccinated against it, confirmation of recent COVID-19 may be harder to prove. In a recently published study of SARS-CoV-2 seroprevalence among pediatric patients in the period between March and July 2022, overall seroprevalence was 86.5% [8]. We want to accent clinical and diagnostical similarities and differences between MIS-C, KD and TSS to help in making a final diagnosis. There are review articles describing the differences and similarities between MIS-C, KD and TSS, but only limited retrospective cohort studies are available comparing these three conditions [6,9,10,11,12,13,14]. Most studies so far have analyzed and compared clinical and laboratory data, but to lesser extent instrumental investigations between those three pathologies. However, one interesting recent review summarizes the available immunological characteristics which MIS-C, TSS and acute rheumatic fever share [15].

Here, we aim to evaluate patients diagnosed with MIS-C, and to compare epidemiological and clinical data, diagnostic findings (including laboratory and instrumental diagnostic investigations) and treatment approaches with those of patients previously diagnosed with KD and TSS.

## 2. Materials and Methods

### 2.1. Patients

This retrospective observational study was conducted at the Children’s Clinical University Hospital (CCUH) in Riga, Latvia. We collected data from all pediatric patients <18 years of age, who met the Centers for Disease Control and Prevention (CDC) case definition for MIS-C, and who presented to CCUH between December 2020 and December 2021, a period when our hospital experienced two MIS-C peaks [16]. We also retrospectively reviewed data from inpatient medical records of patients <18 years of age diagnosed as having KD and TSS at CCUH between December 2015 and December 2021. Patients with these pathologies were selected, using ICD-10 diagnosis codes U10.9, A48.3 and M30.3 in the electronic Latvian health record system “Andromeda”. The information included patient demographics, clinical and laboratory characteristics, treatment and disease outcomes. To determine numbers of KD and TSS cases, patients diagnosed during the pandemic (starting from January 2020) were included if they tested negative for SARS-CoV-2. All the patients with KD met the diagnostic criteria for the disease [17]. The diagnosis of staphylococcal TSS was based on clinical and laboratory criteria [18,19]. The diagnosis of streptococcal TSS was based on clinical criteria and microbiological findings [20]. Laboratory tests first taken after admission to the hospital were analyzed and compared between the research groups. Pathological changes in ECG and echocardiography were noted by analyzing serial tests during the time of hospitalization. Other laboratory and instrumental investigations were performed depending on clinical presentation. As MIS-C often resembled KD at the beginning of presentation, KD criteria were analyzed in all the research groups. The Kobayashi score was used to predict non-responsiveness to IVIG in children with KD [21].

### 2.2. Ethics

The ethics committee of Riga Stradins University and the Institutional Review Board of the Children’s Clinical University Hospital (No. 22-2/450/2021) reviewed and approved this study.

### 2.3. Statistics

Statistical data were analysed using the Statistical Package for the Social Sciences (SPSS) version 26.0 (IBM SPSS Corp.). Statistical significance was evaluated at the level of *p* < 0.05. Differences in clinical and laboratory findings between the study groups were evaluated using Chi-Square or Fisher’s exact test in case of categorical variables, or t-test, ANOVA or the Mann-Whitney U, or the Kruskal Wallis test in the case of continuous variables if they met or had not met the criteria for normal distribution, respectively. Normal distribution of continuous variables was checked using the Kolmogorov-Smirnov test.

## 3. Results

A total of 81 patients were included in this study: 39 (48.1%) with KD (30 with complete and nine with incomplete KD), 29 (35.8%) with MIS-C and 13 (16.1%) with TSS. Pre-existing comorbidities were noted in six (15.4%) patients with KD, three (10.3%) with MIS-C and none with TSS, as shown in Table 1. Among the patients with MIS-C, 16 (55.2%) were male, a proportion similar to that of patients with KD (23; 59%), but among the patients with TSS, eight (61.5%) were female. MIS-C and TSS patients were significantly older than those with KD (*p* < 0.001). Table 2 provides an overview of the patients’ demographics and comorbidities alongside a clinical overview.

The number of days from symptom onset to diagnosis in TSS patients (median, 3 days; interquartile range (IQR), 2.5–5) was less than for KD (median, 7 days; IQR, 5–10) and MIS-C patients (median, 6 days; IQR, 5–7). This may be explained by the fact that patients with TSS had more rapid course of disease with deterioration, therefore they were brought to the hospital sooner than those with KD (*p* = 0.02) and MIS-C (*p* = 0.005). All patients were hospitalized with one dominating symptom—fever. Median duration of fever was longer in KD and MIS-C patients, and statistically significant for less time in TSS patients (*p* < 0.001). Complete KD criteria or incomplete KD criteria were fulfilled in all KD patients, 27 (93.1%) MIS-C patients and only four (30.8%) TSS patients. More than half of patients with MIS-C (51.7%) and over three-fifths of patients with TSS (69.2%) were admitted to an intensive care unit, a higher proportion than for KD (15.4%, *p* < 0.001). Most common indication for admission to ICU was hemodynamic instability and refractory shock. There were no deaths among patients with MIS-C, KD or TSS.

A summary of clinical characteristics is shown in Figure 1 and Appendix A. The clinical manifestations of MIS-C, KD and TSS overlapped substantially. However, compared with KD patients, those with MIS-C were more likely to have gastrointestinal symptoms such as abdominal pain (79.3% vs. 23.1%; *p* << 0.001), diarrhea (31% vs. 12.8%; *p* < 0.004) and both diarrhea and vomiting (34.5% vs. 12.8%; *p* < 0.002), shortness of breath (48.3% vs. 20.5%; *p* < 0.02) and headaches (51.7% vs. 17.9%; *p* < 0.003). They were also more likely to have pleural effusion (75.9% vs. 23.1%; *p* < 0.001) and cardiovascular involvement, including shock (62.1% vs. 17.9%; *p* < 0.001) and hypotension (55.2% vs. 5.1%; *p* < 0.001), valvular insufficiency (72.4% vs. 7.7%; *p* < 0.001), pericardial effusion (41.4% vs. 0%; *p* < 0.001), systolic disfunction (27.6% vs. 0%; *p* < 0.001). All MIS-C patients had cardiac involvement. Most patients had increased levels of high-sensitivity cardiac troponin I and changes on ECG that may indicate inflammatory changes in myocardium. Compared with MIS-C, patients with KD were more likely to have cervical lymphadenopathy (75.8% vs. 41.4%; *p* < 0.006) and arthralgias (23.1% vs. 3.4%; *p* << 0.001). Patients with TSS had lower rates of conjunctival injection (53.8% for TSS patients vs. 89.7% for KD patients; *p* = 0.01), cervical lymphadenopathy (15.4% vs. 75.8%; *p* < 0.001) and musculoskeletal symptoms such as arthralgia (7.7% vs. 23.1%; *p* < 0.007), and synovitis (7.7% vs. 71.8%; *p* < 0.001). All TSS patients had rashes. Table 3 outlines the findings of the instrumental investigations and other diagnostic tests among the study groups. 

We have provided a summary of laboratory findings in Figure 2 and Appendix A. Compared with patients with MIS-C and TSS, KD patients had significantly lower C-reactive protein (CRP) levels, and higher lymphocyte and platelet levels. ESR was significantly lower in patients with TSS compared with those with MIS-C and KD. MIS-C patients had significantly higher levels of fibrinogen, ferritin and cardiac biomarkers (high-sensitivity cardiac troponin I (hs-cTnI), N-terminal (NT)-pro hormone BNP (NT-proBNP)) at the beginning of presentation than was observed in KD and TSS patients. Lower albumin and sodium levels were more commonly observed in TSS and MIS-C patients than those with KD.

Pathological changes on electrocardiograms (ECGs) were detected in almost every MIS-C patient (*n* = 28, 96.6%) during the acute phase of the disease. In most cases, these changes were transient and had disappeared by the time of discharge. In comparison with MIS-C patients, heart rhythm disturbances were seen less frequently in KD (*n* = 12, 30.8%, *p* = 0.003) and TSS patients (*n* = 1, 7.7%, *p* < 0.001). The most prevalent changes seen on the ECGs were non-specific changes in ST segments, intraventricular conduction disturbances and prolonged QTc intervals. A cardiac MRI performed in one MIS-C patient during the acute phase of the disease showed signs of subacute myocarditis. One year later, the same patient had a normal cardiac MRI finding. In 16 other MIS-C patients a cardiac MRI was performed 9–12 months after acute manifestation. So far, 15 of these have been analysed: two revealed signs of myocarditis, and another a decreased left-ventricular myocardial mass index, but in twelve patients pathological changes were not detected.

The first-choice therapy for all children with KD was monotherapy with intravenous immunoglobulin (IVIG). IVIG therapy was used in all MIS-C patients and one TSS patient. Two KD patients received a second dose of IVIG. Median time in days from illness onset and IVIG administration in KD patients was 7 days (IQR 5–10 days) and 6 days (IQR 5–7 days) in MIS-C patients. We analysed patients’ clinical data to calculate potential IVIG resistance according to Kobayashi criteria [21]. Median results were lower in TSS patients (0 points) and KD patients (median 1 point (IQR 1–3 points)) than in MIS-C patients (median 4 points (IQR 2.5–6.0 points; *p* < 0.001)) (*p* < 0.001). The low-risk group for IVIG resistance (0–4 points) comprised 36 (92.3%) KD patients, 15 (51.7%) MIS-C patients and one TSS patient. The high-risk group for IVIG resistance (≥5 points) included 3 (7.7%) KD and 14 (48.3%) MIS-C patients. Glucocorticoids (GCs) were given to 20.5% of KD patients. Seven of the patients received pulse therapy, while one patient took GCs in low doses (2 mg/kg/day). The majority of MIS-C patients (*n* = 29, 96.5%) were treated with GCs, mostly in low doses, but 13.8% (*n* = 4) of patients required glucocorticoid (GC) pulse therapy because the clinical effect of low doses was ineffective. Only one patient with TSS received low-dose GCs. Antimicrobials were prescribed for 92.3% (*n* = 36) of KD patients, 96.6% (*n* = 28) of MIS-C patients and all TSS patients. Acetylsalicylic acid (ASA) was prescribed for 94.9% (*n* = 37) of KD patients and all MIS-C patients. Anticoagulation therapy was completed in 62.1% (*n* = 18) of MIS-C and 23.1% (*n* = 3) of TSS patients. Because of hemodynamic instability and fluid refractory shock, 37.9% (*n* = 11) of MIS-C and 46.2% (*n* = 6) of TSS patients received inotropic agents (adrenaline, noradrenaline and/or milrinone). Additionally, one MIS-C patient received an interleukin-1Ra (Anakinra) because of refractory disease.

## 4. Discussion

Particular attention was given to MIS-C due to the high frequency of life-threatening complications and the challenges of differentiating it from other hyperinflammatory conditions. The present study highlights demographic, clinical and laboratory differences among the three disorders that can assist in establishing a diagnosis and appropriate management of affected children. During the COVID-19 pandemic, MIS-C cases shared clinical features with the conditions KD and TSS [13,14,22]. Here, we present the features of MIS-C associated with COVID-19 and compare them with a historical cohort of patients with KD and TSS. Comparison of these three pathologies shows that, although there are several overlapping features, multiple clinical and investigational findings differ between the groups and help to differentiate these pediatric inflammatory entities. The findings of this research present similar results to previously published studies comparing MIS-C with KD and TSS [12,13,14,23]. Patients with TSS were admitted to hospital earlier than the MIS-C and KD patients. As observed in the study by Kostik et al. conducted in St Petersburg, patients with MIS-C were admitted to hospital earlier than KD patients, possibly due to the family concerns about potential cause of SARS-CoV-2 [23]. In our research, we concluded that the time period from the beginning of symptoms to admission to hospital was similar between the MIS-C and KD study groups. In comparison, patients with TSS were admitted significantly sooner than the other two groups—most likely because of more rapid deterioration of clinical state. We noted that children with MIS-C and TSS tended to be older; indeed, those with KD usually are younger. The mean onset ages of MIS-C and TSS in our study were 9.8 and 11.3 years, respectively, which is comparable with previously published studies [23,24,25,26]. Gender distribution in our research groups corresponds with the data available in the literature [12,13,23]. In comparison with other available sources and articles, MIS-C patients in our research more often fulfilled either classic or incomplete KD criteria (93.1% versus 40–50%) [7]. MIS-C was more commonly associated with abdominal pain, vomiting and diarrhea, shortness of breath, headaches and cardiovascular involvement compared with KD patients. KD patients had the highest rates of conjunctivitis, cervical lymphadenopathy and musculoskeletal symptoms, along with the lowest rates of cardiovascular symptoms such as shock and hypotension, pathological changes in ECGs and pleural effusion. TSS was more commonly associated with shock and hypotension than MIS-C and KD. TSS patients were more likely to have acute kidney injury.

Patients with MIS-C had a higher neutrophil count, more CRP, and ferritin, as well as profound lymphopenia and hyponatremia compared with those with KD and TSS. They also tended to have a lower platelet count, higher fibrinogen and a greater elevation of troponin. Patients with TSS had higher white blood cell, neutrophil and platelet counts and a lower lymphocyte count, with reduced ESR, fibrinogen and troponin levels compared with MIS-C. Thrombocytosis is typical of KD, but less common in MIS-C patients, according to the literature [14,27,28].

There are some differences in the pathological findings of echocardiograms between study groups. Thus, valvular insufficiency, pericardial effusion, systolic dysfunction and myocarditis were more common in MIS-C compared with KD and TSS, while coronary artery changes were more typical of KD, as similarly noted in previously published studies [12,13,23]. After additional investigation analysis, we can conclude that all the patients with suspected MIS-C diagnosis should be carefully investigated for possible cardiovascular involvement, as those patterns significantly differ from the KD and TSS cases.

There are clinical practice recommendations regarding treatment of MIS-C, KD and TSS from several organizations and authors [16,29,30,31,32,33,34]. Overall, our experience showed good outcomes using IVIG, corticosteroids, ASA and anticoagulants.

This study has several limitations. Firstly, our study was retrospective in nature, so clinical and laboratory data may not have been consistently collected from medical records. To particularize, some investigations for a proportion of patients may be missing because of technical issues (results not added to electronic system), or serial laboratory tests had not been performed, because at the time of presentation there were no indications for them (for example, cardiac biomarkers and echocardiography in TSS patients). As a result, performance of different laboratory tests and other investigations was heterogenous among research groups. This may influence the results of comparison. Equally, the number of cases studied was small. Secondly, the present study includes data from just one center. On the other hand, all the patients with MIS-C, Kawasaki disease and toxic shock syndrome from Latvia are treated in the only tertiary children’s hospital (CCUH), therefore data are more comparable and representative for the Latvian population. To improve this research, a summary of results may be helpful for clinicians in everyday use. Thus, Godfred-Cato et al. have prepared four diagnostic scores that may help in distinguishing MIS-C from KD, TSS and COVID-19 [12].

## 5. Conclusions

This study of patients with MIS-C, KD and TSS identified patterns of demographic features, comorbidities, clinical characteristics, laboratory and instrumental investigation findings. These patterns may help differentiate between MIS-C and other hyperinflammatory conditions.

## Figures and Tables

**Figure 1 medicina-59-00626-f001:**
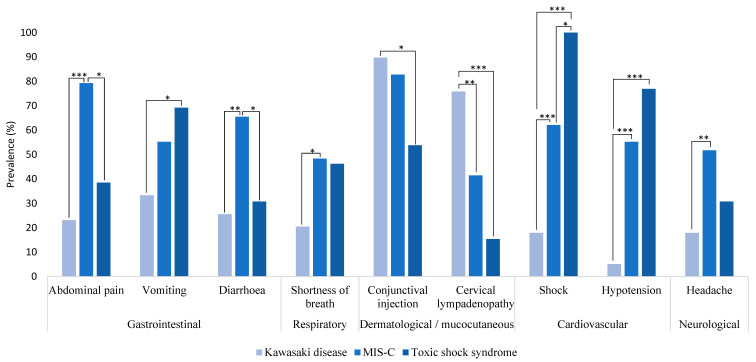
Clinical characteristics of patients with KD, MIS-C and TSS. * (*p* < 0.05), ** (*p* < 0.01), *** (*p* < 0.001).

**Figure 2 medicina-59-00626-f002:**
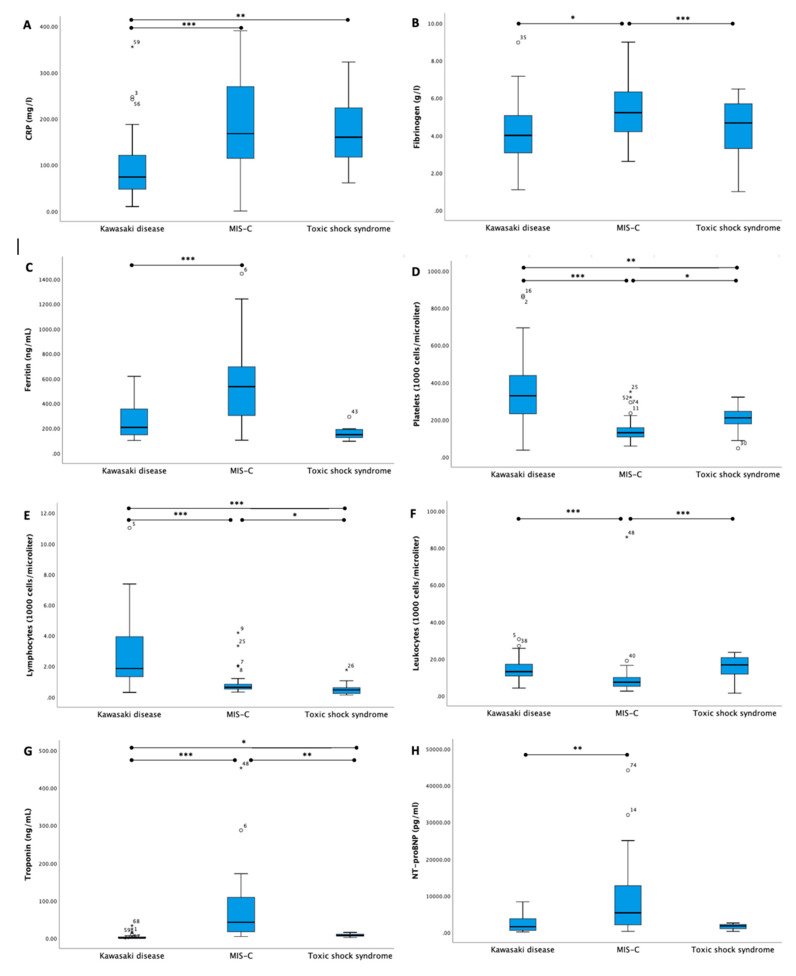
Laboratory markers of patients with MIS-C, KD and TSS (levels of significance * (*p* < 0.05), ** (*p* < 0.01), *** (*p* < 0.001)).

**Table 1 medicina-59-00626-t001:** Pre-existing comorbidities among research groups.

	KD	MIS-C	TSS
Prevalence of comorbidities in groups	6 (15.4%)	3 (10.3%)	0 (0%)
Comorbidities specified	Atrial septal defect Asthma Atopic dermatitis Patent ductus arteriosus Psychomotor development retardation after perinatal hypoxic brain injury Unspecified malabsorption syndrome	Gallstone disease Asthma (remission) Autism spectrum disorder	

**Table 2 medicina-59-00626-t002:** Demographic features, comorbidities and clinical overview of MIS-C, KD and TSS patients.

Variable	KD	MIS-C	TSS	Total	*p*-Value	KD vs. MIS-C, *p*-Value	MIS-C vs. TSS, *p*-Value	KD vs. TSS, *p*-Value
Number of patients, *n* (%)	39 (48.1%)	29 (35.8%)	13 (16.1%)	81 (100%)				
Age in years; mean (SD)	3.9 (3.7)	9.8 (4.5)	11.3 (4.5)	7.2 (5.2)	<0.001	<0.001	0.31	<0.001
Sex								
Female, *n* (%)	16 (41.0)	13 (44.8)	8 (61.5)	37 (45.7)	0.44			
Male, *n* (%)	23 (59.0)	16 (55.2)	5 (38.5)	44 (54.3)				
Comorbidities, *n* (%)	6 (15.4)	3 (10.3)	0	9 (11.1)	0.4			
Day of hospitalization since onset of symptoms, median (IQR)	5.0 (3.0–7.0)	5.0 (4.0–6.0)	3.0 (2.0–5.0)	5.0 (3.0–6.0)	0.03	0.74	0.005	0.02
Time from symptom onset to diagnosis, median (IQR)	7.0 (5.0–10.0)	6.0 (5.0–7.0)	3.0 (2.5–5.0)	6.0 (5.0–9.0)	<0.001	0.03	<0.001	<0.001
Length of stay in hospital, days, median (IQR)	12.0 (9.0–16.0)	12 (11.5–16.0)	9.0 (6.5–12.0)	12.0 (9.5–15.0)	0.02			
Duration of fever in days, median (IQR)	9.0 (7.0–12.0)	7.0 (5.0–8.0)	4.0 (3.0–5.0)	7.0 (5.0–9.0)	<0.001	<0.001	<0.001	<0.001
Fever lasting at least 5 days, *n* (%)	38 (97.4)	27 (93.1)	4 (30.8)	69 (85.2)	<0.001	0.57	<0.001	<0.001
Unfulfilled KD criteria, *n* (%)	0	2 (6,9)	9 (69,2)	11 (13,6)	<0.001	0.10	0.001	<0.001
Classic KD criteria, *n* (%)	30 (76.9)	12 (41.4)	2 (15.4)	44 (54.3)		0.007	0.99	0.27
Incomplete KD criteria, *n* (%)	9 (23.1)	15 (51.7)	2 (15.4)	26 (32.1)		0.53	<0.001	<0.001
Admission to the Pediatric Intensive Care Unit (PICU), *n* (%)	6 (15.4)	15 (51.7)	9 (69.2)	30 (37.0)	<0.001	0.001	0.29	0.001
Days in PICU, median (IQR)	4 (2.5–6)	2 (2–3)	3 (2–3.5)	3 (2–4)	0.19			
Outcome								
Died, *n* (%)	0	0	0	0	-			

**Table 3 medicina-59-00626-t003:** Comparison of findings in instrumental diagnostic investigations and other diagnostic tests across study groups.

Variable	KD	MIS-C	TSS	Total	*p*-Value	KD vs. MIS-C, *p*-Value	MIS-C vs. TSS, *p*-Value	KD Vs. TSS, *p*-Value
Thoracic X-ray Pneumonia	6 (15.4)	10 (34.5)	4 (30.8)	20 (24.7)	0.17			
Thoracic X-ray or ultrasound Pleural effusion	9 (23.1)	22 (75.9)	5 (38.5)	36 (44.4)	<0.001	<0.001	0.04	0.30
Electrocardiogram (ECG) findings								
Long QTc interval, *n* (%)	0	7 (24.1)	0	7 (11.9)	0.02	0.01	0.56	-
Changes in ST segment, *n* (%)	5 (19.2)	26 (89.7)	0	31 (52.5)	<0.001	<0.001	<0.001	0.99
Atrioventricular (AV) conduction disturbances, *n* (%)	1 (3.8)	6 (20.7)	0	7 (11.9)	0.19			
AV dissociation, *n* (%)	0	3 (10.3)	0	3 (5.1)	0.39			
Intraventricular conduction disturbances, *n* (%)	7 (26.9)	16 (55.2)	0	23 (39.0)	0.02	0.03	0.1	0.55
1^st^ degree AV block, *n* (%)	1 (3.8)	2 (6.9)	0	3 (5.1)	0.99			
2^nd^ degree AV block, *n* (%)	0	0	0	0	-			
Atrial extrasystoles, *n* (%)	1 (3.8)	2 (6.9)	0	3 (5.1)	0.99			
Ventricular extrasystoles, *n* (%)	1 (3.8)	1 (3.4)	1 (25.0)	3 (5.1)	0.27			
Echocardiography findings								
Pathological changes, *n* (%)	5 (12.8)	24 (82.8)	0	29 (39.7)	<0.001	<0.001	0.001	0.99
Valvular insufficiency, *n* (%)	3 (7.7)	21 (72.4)	0	24 (29.6)	<0.001	<0.001	<0.001	0.56
Pericardial effusion, *n* (%)	0	12 (41.4)	0	12 (14.8)	<0.001	<0.001	0.008	-
Systolic disfunction, *n* (%)	0	8 (27.6)	0	8 (9.9)	<0.001	0.001	0.04	-
Coronary artery changes in acute phase, *n* (%)	4 (10.3)	1 (3.4)	0	5 (6.2)	0.46			
Coronary artery changes after 1 month, *n* (%)	3 (7.7)	0	-	2 (3.4)	0.53			
Coronary artery changes after 3 months, *n* (%)	3 (8.1)	0	-	3 (5.4)	0.54			

## Data Availability

The raw data supporting the conclusions of this article will be made available upon request to the corresponding author.

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
