# Peer review of "Comparison of Characteristics and Outcomes of Multisystem Inflammatory Syndrome, Kawasaki Disease and Toxic Shock Syndrome in Children"

_medicina, 2023, doi:10.3390/medicina59030626_

Round 1

Reviewer 1 Report

Thank you for the opportunity to review your paper. Here are some thought and suggestions:

General: This retrospective chart review of cases of MIS-C, Kawasaki Disease, and Toxic Shock Syndrome from a single center provides clinical details of children with these diseases, but does not really add to the literature or understanding of these topics. It is unclear what this report and these data add, as the majority of the clinical findings presented are in line with what is already known and part of the case definitions and diagnostic criteria.

Abstract:

·       If there is room in the word count, it is helpful to define all abbreviations when they are first used.

·       It is not clear what the purpose of your study is from the abstract. It is well-known that there are overlapping features of these 3 diseases and that laboratory and additional investigation is needed to make the diagnoses (as these are part of the diagnostic criteria).

·       Consider including the statistical significance in the results included in the abstract.

Introduction:

·       Line 34-40: You can probably skip over the background information about the COVID-19 pandemic as this is universally known at this point.

·       Line 43-45: Strongly suggest actually writing out the definition of MIS-C, and also mentioning that the CDC case definition has been updated since your study ended.

·       Suggest summarizing what is actually currently known about MIS-C in relation to KD and TSS.

·       Strongly suggest to highlight what gaps in the literature currently exist and what gaps you are planning to fill with this analysis. What are you planning to show us that is new or different?

Methods:

·       Line 68-70: This is not clear – please clarify your methods for identifying patients diagnosed with KD and TSS.

·       Please explicitly state how the patients were identified – was this through diagnostic codes in an electronic health record system? How were patients verified as having diagnoses of MIS-C, KD, or TSS, how was the data collected?

Results:

·       Line 100: This sentence is hard to follow – do you mean that 93% of MIS-C patients also met criteria for complete or incomplete KD? Were the KD criteria applied to all patients regardless of the diagnosis that was assigned? This should be explained in your methods.

·       Line 121: This finding of patients with KD having more cervical lymphadenopathy is not surprising, as this is a principal feature of KD and part of the diagnostic criteria. It would be helpful, in the discussion, to put into context which results were expected, because of the diagnostic criteria and case definitions, and which results are interesting, new, or helpful for medical decision making or management.

·       Line 174: Suggest including a reference for the Kobayashi criteria. Also consider including this in the methods and explaining why this was done.

·       Line 179: May not be grammatically correct with the dashes and commas. Consider rephrasing.

·       Line 185: There is a typo – should be ASA

Discussion:

·       Line 192 – the first sentence of your discussion should be a high-level summary or take-home point from your findings, and this sentence is not.

·       Line 202 – hospital is misspelled

·       Line 203 – sentence is hard to read, consider rephrasing

·       In general, the discussion seems to be restating findings presented in the results. It is not helpful to repeat the information, please provide a discussion around why your findings are interesting and important and what they add to the literature and clinical understanding of these diseases.

·       In your limitations section, please explain these further and explain how they may have affected your results. Specifically, how would clinical and laboratory data be missing from your review of the medical records?

Conclusions:

·       Many of the patterns you described are well-known features of these diseases, and are often part of the diagnostic criteria. What did your results add? How might these results be used in practice?

Table 1:

·       Strongly suggest to add a Total row to provide the total N for each column at the top of the table.

·       It appears that there is a typo with a comma instead of a period for the row “No KD criteria”

Author Response

Dear Editor in chief and Reviewers,

On behalf of authors, thank you for reviewing our manuscript entitled “Comparison of Characteristics and Outcomes of  Multisystem Inflammatory Syndrome, Kawasaki Disease and Toxic Shock Syndrome in Children " submitted to Medicina (manuscript ID: medicina-2234982). We sincerely appreciate all the valuable comments, which helped us to improve the quality of the article. Our responses to the Reviewers’ comment are described below in a point-to-point manner. Appropriate changes, suggested by the Reviewers, has been introduced to the manuscript (highlighted). 

The article has been prepared in accordance with Medicina manuscript formatting guidelines. Let me emphasize our full readiness to make any further improvements to the manuscript if needed.

Thereby, I affirm that this manuscript has not been published previously, accepted for publication elsewhere and it is not under consideration for publication elsewhere. All authors have contributed significantly to the work and have reviewed and approved the final version of the manuscript.

We hope that our manuscript will be acceptable for publication in Medicina.

Sincerely,

Lizete Kļaviņa

Reviewer 1 commentary 1:  If there is room in the word count, it is helpful to define all abbreviations when they are first used.

Response to Reviewer commentary 1: Thank you for this suggestion. We have illustrated all abbreviations in the abstract (lines 17-35, page 1).

Reviewer 1 commentary 2:  It is not clear what the purpose of your study is from the abstract. It is well-known that there are overlapping features of these 3 diseases and that laboratory and additional investigation is needed to make the diagnoses (as these are part of the diagnostic criteria)

Response to Reviewer commentary 2: Thank you for pointing this out. We have added the aim of the study and explained the meaning of it:

COVID-19 pandemic has brought a lot of challenges. One of them is multisystem inflammatory syndrome in children (MIS-C) that develops 2-6 weeks after acute SARS-CoV-2 infection. First cases were reported in April 2020, and described clinical presentation similar to incomplete Kawasaki disease (KD) or toxic shock syndrome (TSS). During the pandemic, another diagnostic challenge is, that most of patients have already had COVID-19 infection before or have been vaccinated against it, therefore positive COVID-19 history has minor meaning for diagnosing MIS-C. Our objective is to compare clinical signs and investigations between patients with MIS-C, KD and TSS.

Changes are seen in line 17-24 (page 1).

Reviewer 1 commentary 3:  Consider including the statistical significance in the results included in the abstract.

Response to Reviewer commentary 3: Thank you for your comment. We have added the statistical significance in the results included in the abstract. Changes are seen in lines 32-38, page 1.

In comparison with TSS and KD, patients with MIS-C more often presented with abdominal pain (p<0.001) and diarrhoea (p=0.003). All MIS-C patients had cardiovascular involvement. In comparison with KD and TSS, MIS-C patients had higher levels of ferritin (p<0.001), fibrinogen (p=0.04), cardiac biomarkers’, but lower levels of platelets and lymphocytes (p<0.001). KD patients tended to have lower peak of C-reactive protein (CRP) (p<0.001), but higher levels of platelets. Acute kidney injury was more often observed in TSS patients (p=0.01).

Reviewer 1 commentary 4: You can probably skip over the background information about the COVID-19 pandemic as this is universally known at this point.

Response to Reviewer commentary 4: Thank you for your comment and suggestion.

We have deleted the beginning of the introduction about COVID-19 in children.

Reviewer1 commentary 5: Strongly suggest actually writing out the definition of MIS-C, and also mentioning that the CDC case definition has been updated since your study ended.

Response to Reviewer commentary 5: Thank you for ponting this out. We have added updated case definition. Changes are seen in lines 45-60, page 2.

This condition has later been termed Multisystem Inflammatory Syndrome in children (MIS-C), and has been defined by the CDC (36). According to the updated case definition, affected children or adolescents <21 years old have:

  • fever ≥38.0 C for ≥24 hours, multisystem (≥2) organ involvement (cardiac, renal, respiratory, hematologic, gastrointestinal, dermatologic or neurological) and laboratory features suggesting MIS-C – at least one of the following: elevated C-reactive protein (CRP), erythrocyte sedimentation rate (ESR), fibrinogen, procalcitonin, d-dimer, ferritin, lactic acid dehydrogenase (LDH), or interleukin 6 (IL-6), elevated neutrophils, reduced lymphocytes and low albumin, evidence of clinically severe illness requiring hospitalization
  • AND a history of COVID-19 disease (positive for current or recent SARS-CoV-2 infection by RT-PCR, serology, or antigen test; or exposure to a suspected or confirmed COVID-19 case within the 4 weeks prior to the onset of symptoms).
  • AND other possible diagnoses must be excluded (36).

Reviewer 1 commentary 6: Suggest summarizing what is actually currently known about MIS-C in relation to KD and TSS. Strongly suggest to highlight what gaps in the literature currently exist and what gaps you are planning to fill with this analysis. What are you planning to show us that is new or different?

Response: Thank you for the commentary. There are only some articles analyzing the common and differential signs between MIS-C, KD, and TSS. We want to emphasize that after  3 years since the beginning of COVID-19 pandemic,  most of patients have already had COVID-19 once or even more than once or have been vaccinated, therefore positive COVID-19 history has minor meaning in distinguishing MIS-C from KD and TSS. Publications that we mentioned in our article mostly analyze clinical characteristics and laboratory results, but what we add are results from instrumental diagnostic investigations. Our advantage is that all the patients of our country with MIS-C, TSS and KD were treated in the only tertiary level hospital in Latvia, therefore data are more comparable.

We have improved introduction part to complement with your suggestion.

We want to emphasize that after 3 years since the beginning of COVID-19 pandemic, most of patients have already had COVID-19 once or even more than once or have been vaccinated, therefore positive COVID-19 history has minor meaning in distinguishing MIS-C from KD and TSS. We want to accent clinical and diagnostical similarities and differences between MIS-C, KD and TSS to help making final diagnosis.

Changes are seen in lines 67-72, pages 2-3.

Reviewer 1 commentary 7: Line 68-70: This is not clear – please clarify your methods for identifying patients diagnosed with KD and TSS.

Response: Thank you for pointing this out. We used for that following sources:

  1. McCrindle, B.W.; Rowley, A.H.; Newburger, J.W.; Burns, J.C.; Bolger, A.F.; Gewitz, M.; Baker, A.L.; Jackson, M.A.; Takahashi, M.; Shah, P.B.; et al. Diagnosis, treatment, and long-term management of Kawasaki disease: A scientific statement for health professionals from the American Heart Association. Circulation 2017, 135, e927–e999.
  2. Case definitions for infectious conditions under public health surveillance. Centers for Disease Control and Prevention. MMWR Recomm Rep. 1997 May 2;46(RR-10):1-55. PMID: 9148133.
  3. Centers for Disease Control (CDC). Repeat injuries in an inner city population--Philadelphia, 1987-1988. MMWR Morb Mortal Wkly Rep. 1990 Jan 12;39(1):1-3. Erratum in: MMWR Morb Mortal Wkly Rep 1990 Feb 23;39(7):123. PMID: 2294395.
  4. Defining the group A streptococcal toxic shock syndrome. Rationale and consensus definition. The Working Group on Severe Streptococcal Infections. JAMA. 1993 Jan 20;269(3):390-1. PMID: 8418347.

Reviewer 1 commentary 8: Please explicitly state how the patients were identified – was this through diagnostic codes in an electronic health record system? How were patients verified as having diagnoses of MIS-C, KD, or TSS, how was the data collected?

Response: Thank you for this comment. We have used diagnostic codes (M30.3; U10.9; A48.3) and all data in an electronic Latvian health record system ‘’Andromed’’.

Reviewer 1 commentary 9:  This sentence is hard to follow – do you mean that 93% of MIS-C patients also met criteria for complete or incomplete KD? Were the KD criteria applied to all patients regardless of the diagnosis that was assigned? This should be explained in your methods.

Response: Thank you for the commentary. Yes, we wanted to show that MIS-C can mimic KD and analysed KD criteria in all the research groups (MIS-C, KD and TSS). As MIS-C often resembled KD at the beginning of presentation, we analyzed KD criteria in all the research groups to show, that MIS-C can easily be confused with KD. In this article we wanted to reflect both similarities and differences between MIS-C, KD and TSS. We have improved materials and methods part, to explain the reason of analyzing KD criteria in all the research groups:

 As MIS-C often resembled KD at the beginning of presentation, we analyzed KD criteria in all the research groups.

Changes are seen in lines 94-95, page 3.

Reviewer 1 commentary 10: Line 121: This finding of patients with KD having more cervical lymphadenopathy is not surprising, as this is a principal feature of KD and part of the diagnostic criteria. It would be helpful, in the discussion, to put into context which results were expected, because of the diagnostic criteria and case definitions, and which results are interesting, new, or helpful for medical decision making or management.

Response: Thank you for this comment. We definitely agree with you. Finding cervical lymphadenopathy is not surprising for us as well.  This finding confirms that the included patients were appropriate.

Reviewer 1 commentary 11: Line 174: Suggest including a reference for the Kobayashi criteria. Also consider including this in the methods and explaining why this was done.

Response: Thank you for pointing this out. We have added a reference for the Kobayashi criteria and included this in the methods and explained why this was done (lines 95-96, page 3).

Reviewer 1 commentary 12: May not be grammatically correct with the dashes and commas. Consider rephrasing.

Response: Thank you for the comment. We have rephrased this sentence differently.

Glucocorticoids (GCs) were given to 20.5% of KD patients. Seven of the patients received pulse therapy, while one patient took GCs in low doses (2 mg/kg/day). 

Changes are seen in lines 206-207.

Reviewer 1 commentary 13:  Line 185: There is a typo – should be ASA.

Response: Thank you for pointing this out. We have corrected that.

Reviewer 1 commentary 14: Line 192 – the first sentence of your discussion should be a high-level summary or take-home point from your findings, and this sentence is not.

Response: Thank you for this comment. We have added this sentence in the discussion part (lines 222-224).

Reviewer 1 commentary 15: hospital is misspelled

Response: Thank you for pointing this out.  We have corrected the misspelled word.

Reviewer 1 commentary 16: sentence is hard to read, consider rephrasing.

Response: Thank you for the commentary! We have changed the structure of this sentence.

In our research, we concluded that time period from the beginning of symptoms to admission to hospital was  similar between the MIS-C and KD study groups. 

Changes are seen in lines 233-235.

Reviewer 1 commentary 17: Many of the patterns you described are well-known features of these diseases, and are often part of the diagnostic criteria. What did your results add? How might these results be used in practice?

Response:  Thank you for your comment. Present study highlights clinical and laboratory differences among the 3 disorders that can assist practically in establishing a diagnosis of affected children.

Reviewer 1 commentary 18: Strongly suggest to add a Total row to provide the total N for each column at the top of the table. It appears that there is a typo with a comma instead of a period for the row “No KD criteria”

Response: As suggested by the reviewer, we have added a total row. With “No KD criteria” we meant the patients who did not fulfill neither complete nor incomplete KD criteria. We specified it in the table.

Reviewer 2 Report

the authors should clarify  this statement ( Distinguishing MIS-C from other hyperinflammatory conditions) in the introduction line 48 .

results section :

line 94 . the authors mentioned Pre-existing comorbidities were noted but you had not had mention it , it is better to illustrate the type of comorbidities and if this will affect the comparison between the three different disease .

line 133 , the authors mentioned that Patients denoted as ns (not significant), but I have not seen any ns on the figure .

The summary of laboratory findings in Figure 2  need to be more illustrated .

line 173 the authors mentioned that "We analysed patients’ clinical data to calculate potential IVIG 173 resistance according to Kobayashi criteria" .... you should add reference for the Kobayashi criteria  .

the discussion section has been written well but it would be better if you add more studies related to your research.

Author Response

Dear Editor in chief and Reviewers,

On behalf of authors, thank you for reviewing our manuscript entitled “Comparison of Characteristics and Outcomes of  Multisystem Inflammatory Syndrome, Kawasaki Disease and Toxic Shock Syndrome in Children " submitted to Medicina (manuscript ID: medicina-2234982). We sincerely appreciate all the valuable comments, which helped us to improve the quality of the article. Our responses to the Reviewers’ comment are described below in a point-to-point manner. Appropriate changes, suggested by the Reviewers, has been introduced to the manuscript (highlighted).

The article has been prepared in accordance with Medicina manuscript formatting guidelines. Let me emphasize our full readiness to make any further improvements to the manuscript if needed.

Thereby, I affirm that this manuscript has not been published previously, accepted for publication elsewhere and it is not under consideration for publication elsewhere. All authors have contributed significantly to the work and have reviewed and approved the final version of the manuscript.

We hope that our manuscript will be acceptable for publication in Medicina.

Sincerely,

Lizete Kļaviņa

Authors’ reply to Reviewer 2

Reviewer 2 commentary 1: the authors should clarify  this statement (Distinguishing MIS-C from other hyperinflammatory conditions) in the introduction line 48.

Response to Reviewer commentary 1: Thank you for this comment. We have clarified hyperinflammatory conditions and changed the order of sentences for more successful continuity. Changes have been made as follows:

Lines 61-64 on page 2: Distinguishing MIS-C from other hyperinflammatory conditions such as Kawasaki disease and toxic shock syndrome can be challenging for health care providers. Therefore, displaying the major clinical and laboratory differences of these pathologies is critical to make a proper differential diagnosis.”

Reviewer 2 commentary 2: the authors mentioned Pre-existing comorbidities were noted but you had not had mention it , it is better to illustrate the type of comorbidities and if this will affect the comparison between the three different disease

Response to Reviewer commentary 2: Thank you for pointing this out. We have summarized pre-existing comorbidities and reflected them in Table 1 (Line 119, page 4). We did not include them at first as none of the patients received long-term systemic treatment on a daily basis and it was assessed that none of the comorbidities could affect development or course of disease.

Table 1. Pre-existing comorbidities among research groups

KD

MIS-C

TSS

Prevalence of comorbidities in groups

6 (15.4%)

3 (10.3%)

0 (0%)

Comorbidities specified

1.       Atrial septal defect

2.       Asthma

3.       Atopic dermatitis

4.       Patent ductus arteriosus

5.       Psychomotor development retardation after perinatal hypoxic brain injury

6.       Unspecified malabsorption syndrome

7.       Gallstone disease

8.       Asthma (remission)

9.       Autism spectrum disorder

Reviewer 2 commentary 3: the authors mentioned that Patients denoted as ns (not significant), but I have not seen any ns on the figure

Response to Reviewer commentary 3: Thank you for pointing this out. We have deleted this note at Figure 1.

Reviewer 2 commentary 4: The summary of laboratory findings in Figure 2  need to be more illustrated.

Response to Reviewer commentary 4: Thank you for this suggestion. We have improved the summary of laboratory findings as follows.

 Compared with patients with MIS-C and TSS, KD patients had significantly lower C-reactive protein (CRP) levels, higher lymphocyte and platelet levels. ESR was significantly lower in patients with TSS compared with those with MIS-C and KD. MIS-C patients had significantly higher levels of fibrinogen, ferritin and cardiac biomarkers’ (high-sensitivity cardiac troponin I (hs-cTnI), N-terminal (NT)-pro hormone BNP (NT-proBNP)) at the beginning of presentation than it was observed in KD and TSS patients. Lower albumin and sodium levels were more commonly observed in TSS and MIS-C patients than those with KD.

Changes are seen in lines 170-177, page 11.

Reviewer 2 commentary 5: the authors mentioned that "We analysed patients’ clinical data to calculate potential IVIG 173 resistance according to Kobayashi criteria" .... you should add reference for the Kobayashi criteria.

Response to Reviewer commentary 5: Thank you for pointing this out.

We have added reference to an article that analyses risk scoring systems for intravenous immunoglobulin resistant Kawasaki disease (line 201).

Reviewer 2 commentary 6: the discussion section has been written well but it would be better if you add more studies related to your research.

Response to Reviewer commentary 6: Thank you for this valuable comment.

We have added references to novel articles written related to our research. Data and results were similar to our research.

Round 2

Reviewer 1 Report

Dear authors,

   Thank you for your response and for your edits. I have a few additional suggestions for your consideration:

Abstract: Thank you for adding your objectives and providing more context to your results. Your abstract is still missing a conclusion statement that sums up what is added by this report.

Introduction:

Line 45-60: I actually think it is most helpful to describe the definition you used for MIS-C for your study, and then you can mention (for the reader’s knowledge) that the CDC case definition was updated. The reference you added for the case definition is an archived webpage with the old case definition. I suggest that you do not cite a webpage that has been archived and is not in use. Also whichever reference you do use, suggest actually re-numbering your references in your next version. For your information, here is a current CDC webpage about MIS-C which contains the new definition: https://www.cdc.gov/mis/mis-c/hcp_cstecdc/index.html.

Line 67: Large multicentre studies are required for what?

Line 68-69: Do you have data to suggest that most children in Latvia have already had COVID or have been vaccinated? Suggest including citation for this statement.

Line 69: While it is true that there are higher levels of immunity and antibody-positivity after more time with SARS-CoV-2 circulation and vaccine, the history and timing of the SARS-CoV-2 infection is still very important in the diagnosis of MIS-C. Strongly suggest editing this statement.

Methods:

Line 81-94: Suggest actually describing how you identified patients with the 3 conditions of interest from your health system and how you collected the data for the analyses in the text of your manuscript.

Discussion:

Line 228: Consider rephrasing so you are not comparing ‘patients’ to ‘inflammatory entities.’

Line 229: Consider ‘studies’ instead of ‘researches’

Line 233: This sentence is hard to read – the earlier hospitalization was thought to be caused by family’s concerns about COVID-19?

Line 261: Please fix the ASA abbreviation throughout.

Line 263: Please explain this first limitation more, in the text. Why would there be clinical information not collected consistently from the medical record? How might this have affected your results?

Line 265: Please explain why serial lab results were not available and how this might have affected your results.

In general, the discussion still does not clearly state which of your findings were expected, based on what is known about the pathophysiology of the diseases and prior published literature, and what is new and noteworthy. How will these findings help clinicians make diagnoses and choose appropriate management?

Author Response

Dear Editor in chief and Reviewers,

On behalf of authors, thank you for reviewing our revised manuscript entitled “Comparison of Characteristics and Outcomes of  Multisystem Inflammatory Syndrome, Kawasaki Disease and Toxic Shock Syndrome in Children " submitted to Medicina (manuscript ID: medicina-2234982). We sincerely appreciate all the valuable comments, which helped us to improve the quality of the article. Our responses to the Reviewers’ comment are described below in a point-to-point manner. Appropriate changes, suggested by the Reviewers, has been introduced to the manuscript (highlighted).

We hope that our manuscript will be acceptable for publication in Medicina.

Sincerely,

Lizete Kļaviņa

Authors’ reply to Reviewer 1

Reviewer 1 commentary 1:  Thank you for adding your objectives and providing more context to your results. Your abstract is still missing a conclusion statement that sums up what is added by this report.

Response to Reviewer commentary 1: Thank you for your commentary.

We have improved results and conclusions in abstract. Researches so far have analysed clinical and laboratory data, but in lesser extent – instrumental investigations. We want to emphasize laboratory and instrumental investigations that should be performed in order to differentiate MIS-C from KD and TSS. Changes are seen from line 16 to line 39.

Reviewer 1 commentary 2: Line 45-60: I actually think it is most helpful to describe the definition you used for MIS-C for your study, and then you can mention (for the reader’s knowledge) that the CDC case definition was updated. The reference you added for the case definition is an archived webpage with the old case definition. I suggest that you do not cite a webpage that has been archived and is not in use. Also whichever reference you do use, suggest actually re-numbering your references in your next version. For your information, here is a current CDC webpage about MIS-C which contains the new definition: https://www.cdc.gov/mis/mis-c/hcp_cstecdc/index.html.

Response to Reviewer commentary 2: Thank you for your commentary. We have corrected this paragraph, and changes are seen from line 46 to line 61.

This condition has later been termed Multisystem Inflammatory Syndrome in chil-dren (MIS-C), and has been defined by the CDC (36). According to the case definition, affected children or adolescents <21 years old have fever ≥38.0 C at least 24 hours, multisystem (≥2 organ) involvement (cardiac, renal, respiratory, hematologic, gastro-intestinal, dermatologic or neurological) and at least one laboratory feature suggesting MIS-C, evidence of clinically severe illness requiring hospitalization and a history of COVID-19 disease (positive for current or recent SARS-CoV-2 infection by RT-PCR, serology, or antigen test; or exposure to a suspected or confirmed COVID-19 case within the 4 weeks prior to the onset of symptoms). Other possible diagnoses must be excluded (36). This case definition was used in this research for MIS-C patient selec-tion, but since January 2023 case definition has been updated: clinical criteria are specified, requesting CRP level ≥30 mg/l, at least two confirmed signs or organ in-volvement from the following: cardiac involvement, mucocutaneous involvement, shock, gastrointestinal involvement and hematologic involvement. Linkage to COVID-19 (laboratory approved or epidemiological data) is now estimated 60 days prior hospitalization (40).

Reviewer 1 commentary 3: Line 67: Large multicentre studies are required for what?

Response to Reviewer commentary 3: Thank you for your question. We have corrected this statement, and added additional comment (lines 95-97).

Most studies so far have analyzed and compared clinical and laboratory data, but in lesser extent – instrumental investigations between those three pathologies.

Reviewer 1 commentary 4: Line 68-69: Do you have data to suggest that most children in Latvia have already had COVID or have been vaccinated? Suggest including citation for this statement.

Response to Reviewer commentary 4: Thank you for your commentary. We have citated recent study from Latvia about seroprevalence of SARS-CoV-2 in pediatric population. Changes are seen in lines 91-93.

Reviewer 1 commentary 5: Line 69: While it is true that there are higher levels of immunity and antibody-positivity after more time with SARS-CoV-2 circulation and vaccine, the history and timing of the SARS-CoV-2 infection is still very important in the diagnosis of MIS-C. Strongly suggest editing this statement.

Response to Reviewer commentary 5: Thank you for your commentary. We have edited this statement. Changes are seen in lines 87-91.

Notably, given the clinical overlap with KD, for many MIS-C patients the only differentiating feature may be the evidence of prior COVID-19 infection. However, as the pandemic has progressed and a greater proportion of the population has had prior SARS-CoV-2 infection and/or been vaccinated against it, confirmation of recent COVID-19 may be harder to prove.

Reviewer 1 commentary 6: Line 81-94: Suggest actually describing how you identified patients with the 3 conditions of interest from your health system and how you collected the data for the analyses in the text of your manuscript.

Response to Reviewer commentary 6: Thank you for your commentary. We have supplemented Materials and Methods according your suggestions. Changes are seen in lines 115-130.

Reviewer 1 commentary 7: Line 228: Consider rephrasing so you are not comparing ‘patients’ to ‘inflammatory entities.’

Response to Reviewer commentary 7: Thank you for your commentary. We have changed this sentence. Changes are seen in lines 265-267.

Comparison of these three pathologies shows that although there are several overlapping features, multiple clinical and investigational findings differ between the groups and help to differentiate these pediatric inflammatory entities.

Reviewer 1 commentary 8: Line 229: Consider ‘studies’ instead of ‘researches’

Response to Reviewer commentary 8: Thank you for your commentary. We have corrected the sentence.

Reviewer 1 commentary 9: Line 233: This sentence is hard to read – the earlier hospitalization was thought to be caused by family’s concerns about COVID-19?

Response to Reviewer commentary 9: Thank you for your commentary. We have changed the sentence. Changes are seen in line 272.

As observed in the study by Kostik et al. conducted in St Petersburg, patients with MIS-C were admitted to hospital earlier than KD patients, possibly due to the family concerns about potential cause of SARS-CoV-2.

Reviewer 1 commentary 10: Line 261: Please fix the ASA abbreviation throughout.

Response to Reviewer commentary 10: Thank you for your notice. We have corrected the abbreviation. Changes are seen in lines 250 and 304.

Reviewer 1 commentary 11: Line 263: Please explain this first limitation more, in the text. Why would there be clinical information not collected consistently from the medical record? How might this have affected your results?

Response to Reviewer commentary 11: Thank you for your commentary. We have clarified this statement. Changes are seen in lines 307-312.

To particularize - some investigations for a part of patients may be missing because of technical issues (result not added to electronical system), some tests have not been performed because at the time of presentation there were no indications for that (for example echocardiography in TSS patients). As a result, performance of different laboratory tests and other investigations was heterogenous among research groups.

Reviewer 1 commentary 12: In general, the discussion still does not clearly state which of your findings were expected, based on what is known about the pathophysiology of the diseases and prior published literature, and what is new and noteworthy. How will these findings help clinicians make diagnoses and choose appropriate management?

Response to Reviewer commentary 12: Thank you for your commentary. We have improved discussion part and added some statements according recommendations.
